# Role of p53 in Regulating Radiation Responses

**DOI:** 10.3390/life12071099

**Published:** 2022-07-21

**Authors:** Ryuji Okazaki

**Affiliations:** Department of Radiobiology and Hygiene Management, Institute of Industrial Ecological Sciences, University of Occupational and Environmental Health, Japan, Kitakyushu 807-8555, Japan; ryuji-o@med.uoeh-u.ac.jp; Tel.: +81-93-691-7549

**Keywords:** p53, radiation, carcinogenesis, adaptive response

## Abstract

p53 is known as the guardian of the genome and plays various roles in DNA damage and cancer suppression. The *p53* gene was found to express multiple p53 splice variants (isoforms) in a physiological, tissue-dependent manner. The various genes that up- and down-regulated p53 are involved in cell viability, senescence, inflammation, and carcinogenesis. Moreover, p53 affects the radioadaptive response. Given that several studies have already been published on p53, this review presents its role in the response to gamma irradiation by interacting with MDM2, NF-κB, and miRNA, as well as in the inflammation processes, senescence, carcinogenesis, and radiation adaptive responses. Finally, the potential of p53 as a biomarker is discussed.

## 1. Introduction

In 1979, p53 was identified by Arnold J. Levin et al. as a protein that binds to the large T antigen of the oncovirus simian virus (SV40) [1]. Almost at the same time, p53 was reported by Kress et al. [2] and Lane et al. [3]. In 1984, p53 was reported to be an oncogene [4,5,6,7]; however, this was a function of a mutant form of p53 found in cancer cells. In 1989, the *p53* gene was reported to be a tumor suppressor gene [8,9]. Thus, p53 was called “The Guardian of the Genome” [10]. Ataxia telangiectasia mutated (ATM) and ataxia telangiectasia and Rad3-related kinase (ATR) exist upstream of p53. In response to DNA strand breaks caused by ionizing radiation, ATM is activated by chromatin structural changes [11]. Following DNA damage by UV irradiation, ATR is activated by the interruption of DNA replication and transcription [12]. The *p53* gene has a wide range of activities and induces the expresses of genes involved in DNA repair (ribonucleotide reductase regulatory subunit M2 (RRM2), ERCC excision repair 5 (ERCC5), FA complementation group C (FANCC), XPC complex subunit (XPC), X-ray repair cross complementing 5 (XRCC5), growth arrest and DNA damage inducible alpha (GADD45A), Pre-intermoult gene 1 (Pig1), etc.), cell cycle arrest (p21, 14-3-3σ, Riprimo, GADD45A, miR34, cyclin dependent kinase 2 (CDK2), cyclin E1 (CCNE1), etc.), apoptosis (GADD45A, BCL2 associated X (BAX), PIG3, p53 upregulated modulator of apoptosis (Puma), NADPH oxidase activator (NOXA; now known as phorbol-12-myristate-13-acetate-induced protein 1 [PMAIP1])), tumor necrosis factor (TNF), Killer/Dr5, FAS (known as cluster of differentiation 95 (CD95)), p53Alp, apoptotic peptidase activating factor 1 (APAF1), prolyl endopeptidase (PREP), p53-inducible death-domain-containing protein (PIDD), DNA damage-regulated autophagy modulator (DRAM), Fas ligand (FASL), mir34c, p53-regulated apoptosis-inducing protein 1 (tpip1), etc.), autophagy (DRAM, Sestrin 1, Sestrin 2, etc.), antioxidant function (Sestrin 1, Sestrin 2, glutathione peroxidase 1 (GPX1), etc.), senescence (resistance to audiogenic seizures (RAS), MYC proto-oncogene (MYC), PML nuclear body scaffold (PML), plasminogen activator inhibitor-1 (PAI1), p21,Genes of the SASP, mir34a, etc.), and heat shock protein (HSP70) [13,14,15,16,17,18,19,20,21].

The *p53* gene accumulates in the cell nucleus after radiation exposure [22,23,24]. The expression of the *p53* gene was shown to be induced by at least 10 mGy of total body irradiation of mice [21]. Currently, a search for “p53” reveals more than 100,000 articles, with more than 5000 reports per year since 2011. A search of “p53” and “radiation” resulted in more than 8000 papers, and approximately 300 papers have been published every year for the past 20 years. In this review, the role of p53 after ionizing irradiation will be presented with the latest data.

## 2. p53 Isoforms and Their Responses Response to Radiation

The *TP53* gene is composed of 11 exons [25,26], the end-products of alternative usage of promoters (P1 and P2) and splicing of intron-2 and intron-9 (Figure 1A). The p53 protein comprises 393 amino acids and is classified into six domains (Figure 1B) [25,27]: (1) transcriptional activation domain (TAD) (residues 1–67), which can be classified into TAD I (residues 1–40) and TAD II (residues 41–67); (2) interaction with various proteins, including the proline-rich region (residues 67–98); (3) conservation of most p53, i.e., the central core domain (residues 98–303); (4) DNA-binding domain with >90% of p53 mutations causing cancer in humans, i.e., nuclear localization signal (residues 303–323); (5) tetramerization domain (residues 323–363); (6) C-terminal basic domain (residues 363–393) i.e., a nonspecific DNA-binding domain that recognizes and binds damaged DNA [25,27]. Twelve p53 protein isoforms, p53α, p53β, p53γ, Δ40p53α, Δ40p53β, Δ40p53γ, Δ133p53α, Δ133p53β, Δ133p53γ, Δ160p53α, Δ160p53 β, and Δ160p53γ have been identified. Depending on the cell type, *p53* mRNA translation starts at different codons. Then, the mRNA transcribed from the proximal promoter (P1) initiates codon 1 and/or 40 translation, whereas that transcribed from the internal promoter (P2) initiates codon 133 and/or 160 translation. p63 and p73 are family members of p53 [28,29].

By irradiation, the p53 isoform Δ113p53 (as Δ133p53 in human) is a p53 target gene that antagonizes the apoptotic activity of p53 via activation of *bcl2L* (as *Bcl-x_L_* in human) in zebrafish [30]. Restoration of ∆133p53 expression protects astrocytes from radiation-induced senescence, promotes DNA repair, and inhibits astrocyte-mediated neuroinflammation [31]. Different radiation doses in germ cells during Drosophila oogenesis play different roles depending on p53 isoforms (p53A and p53B) [32]. In the absence of stress, p53A is expressed to support normal oogenesis, and, in the early stages of oogenesis, p53A and p53B are involved in radiation-induced apoptosis, and both isoforms were involved in restoring the number of germline stem cells after high dose of irradiation [32]. p53A, but not p53B, was involved in the maintenance of germline stem cells’ function after high doses of irradiation [32]. It had been known that p63, but not p53, is involved in the irradiation-mediated response in human oocytes [33].

## 3. The Murine Double Minute 2 (MDM2)-p53 Pathway after Radiation

Some X-ray-transformed foci showed amplification and/or overexpression of MDM2, and other foci expressed mutant p53 [34] (Figure 2). In the ML-1 human myeloid leukemia cell line, the p53-regulated genes, namely cyclin-dependent kinase inhibitor 1A (CDKN1A, also known as: CDKN1, p21, WAF1, p21CIP1), GADD45A, and MDM2, were reported to be induced at doses between 2 and 50 cGy [35]. When haploinsufficiency is used in *Mdm2*- and *Mdm4*-lacking mice, the *p53* gene function is activated in response to radiation-induced DNA damage in a p53-dependent manner [36]. Ionizing radiation phosphorylated Thr18 in wild-type (wt) p53-containing cancer cell lines, accompanied by an increased expression of p21 and MDM2. However, ultraviolet (UV) treatment showed a cell line-dependent effect [37]. It has been reported that RPS27, a ribosomal protein, and RPS27L, an RPS27-like protein, regulate DNA damage caused by radiation and other factors in the p53–MDM2 axis [38]. In a study of normal lips and actinic cheilitis (AC), a precancerous lesion, MDM2, and p21 were significantly correlated with p53 in normal lips; however, only MDM2 was correlated with p53 expression in AC. In this report, it was found that p53 was the only predictor of AC and a potential marker of the early malignancy of the lips [39]. In a study of cervical cancer radiotherapy, p53 and MDM2 were found in the remaining cancer cells after treatment, and DNA-dependent protein kinase (DNAPK), which is involved in DNA repair, was activated, resulting in radioresistance [40].

## 4. The p53 and Nuclear Factor-KappaB (NF-κB) Pathway after Radiation

The regulation of the DNA damage response and immune response involves the activity of transcription factors such as p53 protein, activated protein 1 (AP-1), and NF-κB [41,42] (Figure 3). In mouse lymphoid cells and thymocytes, the activation of NF-κB and apoptosis after X-irradiation is p53 dependent [43]. When DNA damage occurs, ATM activates p53 in the cell nucleus, which in turn activates the NF-κB pathway via the cytoplasmic IκB complex (IKKα, IKKβ, and NEMO) [44]. Transcription of NF-κB and NF-κB target genes was shown to be involved in radioresistance in human keratinocytes in which TP53 is inactivated by fractionated ionizing radiation (2 Gy/fraction; total dose, 60 Gy) [45]. Even in p53-deficient tumor cells, the fungal metabolite gliotoxin inhibited NF-κB activation with high selectivity, activated c-Jun N-terminal kinase (JNK), released cytochrome c from mitochondria, and potently stimulated the caspase cascade, inducing the cleavage of caspase-9, -8, -7, and -3 [46]. Inhibiting NF-κB activation is a strategy to protect against radiotherapy in cancer. Inhibition of NF-κB signaling may trigger oncogenesis independent of initiating mutations in the Ha-ras gene or additional mutations in the *p53* gene in skin cancer induced by gamma irradiation [47]. It has been reported that, in human umbilical vein endothelial cells (HUVECs), the protein levels of the mitogen-activated protein kinase (MAPK) pathway molecules, p38, p53, p21, and p27, were increased after gamma irradiation, and that the activation of the NF-κB pathway was also induced [48]. The PIDD is involved in caspase-2 activation and apoptosis in response to DNA damage and has an important role in NF-κB activation [49]. Irradiation was shown to activate or repress NF-κB and induce differential expression downstream genes depending on the expression status of the *p53* gene in the cells [50]. NF-κB is a negative regulator of autophagy in mutant p53 (p53-R273H) cells [51]. p300 or p65 inhibition not only activated autophagy but also induced radiosensitivity in p53-R273H cells [51]. This finding indicates that autophagy may be regulated by NF-κB [51]. When mice were irradiated at a young age, p53 and NF-κB were activated at an older age, but not at a middle-aged age [52]. This finding indicates that exposure at a young age accelerated the senescence process [52]. Furthermore, attenuation of the p53 function promoted inflammation and carcinogenesis [53], and NF-κB activity plays a major role in this process [54,55,56]. If gamma irradiation (activating the p53 pathway) precedes TNF (activating the NF-κB pathway), the role of NF-κB is proapoptotic; if the stimuli are reversed, the role of NF-κB is antiapoptotic [57]. Radiation-induced p53 and NF-κB can be dependent or independent.

## 5. p53 and MicroRNAs (miRs) after Radiation

MicroRNAs (miRs) are 21–25 base-long, single-stranded RNA molecules that are involved in the post-transcriptional regulation of gene expression, cell division, differentiation, apoptosis, metabolism, and tumorigenesis in eukaryotes [58,59,60]. Radiation-induced DNA damage induces the expression of miR-34 by the *p53* gene [61,62], and overexpression of miR-34b increases radiosensitivity [63] (Figure 4). MiR-34a remains stable in serum after irradiation and may be a new indicator of radiation injury, strongly suggesting that miR-34a may be a new indicator, mediator, and target of radiation injury, radiosensitivity, and radiation protection [64]. MCF-7 cells have higher γ-ray-induced p53 expression than HeLa cells; however, the induction of miR-34 shows the opposite pattern, suggesting that miR-34 induction may be cell type-specific [65]. In glioblastoma cells, after 60 Gy irradiation, miR-34a was highly expressed; however, p53 expression was decreased [66]. It is possible that miR-34a regulation induces apoptosis even in cells that have become radioresistant [66]. In human mammary epithelial cells, p38 MAPK-mediated p53 ser15 phosphorylation is essential for radiation-induced functional regulation of miR-34a transcription, and histone modifications also play an important role in miR-34a expression [67].

The expression of let-7a and let-7b, which are downstream of p53, was downregulated by radiation and correlated with changes in the expression of proteins in the p53-regulated apoptosis-promoting signaling pathway, which are important for radiation-induced cytotoxicity [68]. In other words, the p53 direct binds and activates genes, but let-7a and let-7b were repressed by p53. In prostate cancer cells, fractionated irradiation increased the expression of tumor suppressive miR-34a and let-7, which are not dependent on p53 alone [69].

MiR-295 is found at a site identified as a direct transcriptional target of p53 that binds to Bcl2 and may act to suppress apoptosis in tumor cells, making them radioresistant [70]. MiR-155 has been suggested to modulate radiation-induced aging by regulating the expression of tumor protein 53-induced nuclear protein 1 (TP53INP1) downstream of the p53 and p38 MAPK pathways [71]. MiR-375 expression was elevated in recurrent gastric cancer, and the overexpression of miR-375 resulted in the decreased expression of p53 protein [72]. In other words, miR-375 inhibits cell cycle arrest and apoptosis, making the cells radioresistant [72]. MiR-124 expression is induced by p53, and miR-124 suppresses NF-κB p65, leading to the suppression of MYC/BCL2 expression and cell survival in B-cell lymphoma [73]. MiR-17-5p was shown to cause p53-dependent apoptosis and contributes to radiosensitivity in irradiated betel nut chewing-oral squamous cell carcinoma OC3 cells [74]. Increased miR-372 activated p53 via suppression of the PDZ-binding kinase and increased radiosensitivity of tumor cells [75]. MiR-621 is also involved in the radiosensitivity of hepatocellular carcinoma through the p53 signaling pathway [76]. MiR-1246 targets and suppresses the *p53* gene, which has anticancer effects on bladder cancer cells [77]. MiR-30e expression was upregulated in a p53-dependent manner after gamma-irradiation and strongly accelerated and increased the aging phenotype [78]. MiR-214-3p suppressed ATM/P53/P21 signaling when extracellular vehicles (EVs) derived from mesenchymal stem cells (MSCs) suppressed radiation-induced lung injury, resulting in senescence-associated secretory phenotype development [79]. The regulation of p53 by miRNAs has been reviewed previously [80].

## 6. p53 and Inflammation Induced by Radiation

Radiation exposure has been known to produce various immune molecules, such as cytokines, danger-associated molecular patterns (DAMPs), and cellular factors that cause inflammation [81]. The role of p53 in innate immunity and inflammatory responses has already been well established [82,83,84]. In CBA/Ca male mice, at post-irradiation with 4 Gy gamma rays, many NF-κB-positive cells were still present after 100 days, suggesting a persistent inflammation [85]. In the hematopoietic system of irradiated CBA/Ca mice, the microenvironment, a non-target of radiation, is enhanced by effect changes and inflammatory-type signals [86,87,88,89], namely, changes in oxygen- and nitrogen-free radical activity and cytokines such as TNFα and Fas-L, and cyclooxygenase (COX)-2 signaling, are observed [86,87,88,89]. Changes are thought to induce p53-mediated apoptosis [90]. The addition of NS-398, a nonsteroidal anti-inflammatory drug that selectively inhibits COX-2, significantly reduces cell death [91]. In irradiated mouse bone marrow cells, *p53*-gene-dependent apoptosis involves inflammatory mechanisms at 24 h during the apoptotic process [92]. Then, the codon 72 polymorphism in p53 has been reported to affect p53-mediated inflammatory responses at post-irradiation [93]. Macrophage activation after a radiation exposure is caused by radiation-induced apoptosis and may represent a direct p53-mediated pathway of macrophage activation [94]. In human monocytes and macrophages, inflammatory cytokine gene expression may be directly induced by p53- and ATM-dependent mechanisms [81]. p21 deletion in macrophages decreased macrophage inflammatory protein (MIP)-1, MIP-2, IL-1α, and IL-1β expressions, which are involved in the inflammation process [95]. Fetal radiation on day 12 (post-conception) releases inflammatory cytokines into the amniotic fluid on day 16 and subsequently induces *p53*-gene-dependent, apoptosis-related genes, resulting in malformations [96]. Irradiation of primary human retinal microvascular endothelial cells (RECs) rapidly induces the p38 mitogen-activated protein kinase (MAPK) stress kinase-mediated pathway and its downstream effectors (tumor suppressor, p53, intercellular adhesion molecule 1 (ICAM-1)), resulting in inflammatory responses [97]. In the radiation-induced brain injury mice, IκB-α expression was downregulated, accompanied by the upregulated NF-κB expression essential modulator (NEMO) and regulated auto-proteolysis of PIDD [98]. Irradiation activates *cathepsin K (CSTK)*, *nuclear factor of activated T-cells 5 (NFAT)*, *Tartrate-Resistant Acid Phosphatase-5b (TRAP-5b)*, *receptor activator of nuclear factor-kappa B ligand/Osteoprotegerin (Rank l/OPG)*, *IL-1*, *IL-6*, *and TNFα* genes in osteoclasts, induces inflammation and promotes osteoporosis [99]. Irradiation of patient-derived glioblastoma cells induces p53-dependent inflammation through NF-κB signaling [100]. In patients with glioblastoma undergoing radiation therapy, decreased Ki67 labeling index, O-methyl guanine methyl transferase (MGMT) methylation status, isocitrate dehydrogenase (IDH) mutation status, and absence of p53 overexpression have all also been associated with long-term survivors [101]. Radiation-induced pneumonitis in patients with lung cancer treated with radiotherapy is associated with dysregulation of p53 signaling by p53 and ATM polymorphisms [102], which is consistent with the association between miRNAs and p53 signaling during whole-thorax lung irradiation in mice [103]. In patients with Alzheimer’s disease treated with low doses of radiation, *Helicobacter pylori* increased the intestinal permeability through p53-dependent activation of the TLR4/Myd88 inflammatory pathway, causing metabolic dysfunction [104].

Recently, inflammation has been hypothesized to cause carcinogenesis [105]. Chronic inflammation is a major cancer predisposition factor [106,107,108,109]. Irradiation of mice lacking the *p53* gene caused TGF-β-associated inflammation, which was robustly associated with claudin-low tumors [110]. Colorectal cancer in Mlh1-deficient mice is more likely to develop when exposed to radiation and simultaneously induced inflammation [111]. Naringin, a citrus flavonoid, suppresses radiation-induced cytotoxicity of the p53 pathway and inflammation in the NF-κB pathway [112], which is important in suppressing various radiation-induced cytotoxicity. In prostate cancer radiotherapy, transcript levels were elevated, primarily of DNA damage binding protein 2, p21, collagen, laminin, and integrins, as well as the upregulated p53 pathway [113]. Interstitial remodeling, extracellular matrix proteins, and focal adhesion pathways were also strongly upregulated, as was inflammation [113]. These indicate a clustering of apoptosis and programmed cell death, extracellular matrix organization, and inherent changes in immune regulation [113]. Deletion of lysine (K)-specific demethylase 6A (KDM6A) cooperates with p53 haploinsufficiency to promote the activation of the cytokine–chemokine pathway and M2 macrophages polarization, ultimately causing bladder cancer [114].

## 7. p53 and Radiation Induced Carcinogenesis

Loss or mutation of p53 is known to be involved in radiation-induced carcinogenesis [115,116,117]. Radiation-induced mutations at the thymidine kinase locus were 30-fold more common in p53 mutant cells than in p53 wild-type cells, and loss of heterozygosity was more common [118]. Secondary sarcomas arising in the irradiated region of the primary tumor after radiotherapy showed somatic inactivating mutations in one allele of TP53 and were systematically associated with defects in the other allele [119]. Furthermore, induction of p16 has a stronger relationship to carcinogenesis than loss of p53 [120]. Mutations in p53 may not be directly induced by radiation but may arise in cells several generations after irradiation [121]. We have reported that changes in T-cell surface markers occur in a delayed manner when mice are exposed to whole body irradiation, and this phenomenon was observed earlier in p53 heterozygotes [122]. X-irradiated cells undergo delayed apoptosis, which may not be a result of nonfunctioning of the acute TP53 damage response pathway but rather X-ray-induced genomic instability that occurs in the remote descendants of irradiated cells [123].

Hepatocellular carcinoma caused by alpha particles has been related to point mutation at codon 249 of p53 [124,125,126]. Electron and neon ion beam-induced rat skin cancers, but not alpha-induced liver cancer, showed alterations in critical binding regions in exons 5–8 of the *p53* gene [124,127]. Structural genetic abnormalities, including mutations in *p53* (exons 5–8), *RAS* (codons 12, 13, 61), and *Gsα* (codons 201 and 227), and, less frequently, in receptor tyrosine kinases such as RET and NTRK1, have been found in various sporadic thyroid tumors in adults [128]; however, no *p53* gene mutations have been found in thyroid cancer from the Chernobyl nuclear accident [129,130]. In human colorectal cancer, tripartite motif containing 23 has been reported to physically bind p53 and promote p53 ubiquitination, thereby promoting tumor growth [131].

In mice, mTOR signaling is activated through the p53-Fbxw7 pathway early after irradiation, and the suppression of Fbxw7 and mTOR suppressed carcinogenesis [132]. Whole-body irradiation of *p53* wild-type mice resulted in an acute DNA damage response that activated p53 in the bone marrow; however, cells from the bone marrow have a reduced ability to compete with tumor-forming cells in the thymus and promote radiation-induced lymphoma in the thymus via noncell autonomous mechanisms [133]. Although p53 suppresses the development of spontaneous tumors expressing Kras, in the context of exposure to ionizing radiation, regardless of low-LET or high-LET, extra copies of p53 may conversely not protect against radiation-induced lymphoma and may promote Kras mutant lung cancer [134]. In *Mdm2-S394A* mice (a mouse model in which ATM phosphorylation at serine residue 394 of *Mdm2* is abolished), the *Mdm2* response was that the remains destabilized after irradiation, eliminating the need for p53 mutations in Myc-driven carcinogenesis [135]. Akt, a DNA damage effector kinase, phosphorylates Mdm2 and regulates the p53 response to oxidative stress without affecting the p53-mediated effects of ionizing radiation in mice [136]. The mouse model of radiation-induced thymic lymphoma is a well-known model. In p53 wild-type mice, genetic inhibition of Notch1 signaling suppressed the formation of radiation-induced thymic lymphoma and plays an important role in promoting multistep carcinogenesis of thymic lymphoma [137].

## 8. p53 and Senescence Induced by Radiation in Normal Cells

A hallmark of senescence is that it causes a steady arrest of cell proliferation [138]. It is also important in stopping dysfunctional cells’ proliferation, and two tumor suppressor pathways, namely, p53 [138,139,140] and p16/Rb [138,141], have important roles. Alternatively, there is an MDM2-mediated hypoxia-inducible factor 1α (HIF1α) senescence is found, which is not mediated by the *p53* gene [142]. Irradiated bone marrow-derived mesenchymal stem cells were observed to exhibit aging phenotypes, such as elevated levels of aging-related genes p53/p21 but without changes in p16 [143]. In normal human embryo cells, X-irradiation caused sustained TP53 phosphorylation at Ser15 and TP53 protein accumulation, resulting in the induction of CDKN1A (p21 (Waf1/Cip1)) and CDKN2A (p16), which was preceded by the expression of senescence-associated-β-gal [144]. p16, with or without p53, causes senescence in mice after prolonged irradiation [145]. After an ionizing irradiation of normal human fibroblasts, the same distribution of foci containing p53 phosphorylated at serine 15 along with large foci containing phosphorylated ATM and γ-H2AX also cause senescence-like growth arrest, indicative of 53-dependent irreversible G1 arrest [146]. However, telomere shortening is not necessarily required for radiation-induced senescence-like growth arrest [144,146,147]. Although telomere shortening involves the activation of the pRB and p53 pathways, the absence of telomere instability and the defects in the G2 chromosome repair in patients with cancer aged over 80 years of age irradiated with 50 Gy or more of gamma radiation correlates with longevity [148]. Exposure of mouse bone marrow cells to 4 Gy of ionizing radiation induces hematopoietic cell senescence in a p53–p21(Cip1/Waf1)-dependent manner [149]. Continuous irradiation of primary human umbilical vein endothelial cells (HUVECs) with low dose rate gamma radiation rate (4.1 mGy/h) results in senescence via the p53/p21 pathway [150]. Radiation-induced senescence in HUVECs also involves nuclear factor kappa B essential modulator (NEMO) [151]. Suppression of the human papillomavirus (HPV) type 18 E6/E7 gene in HeLa cells by bovine papillomavirus E2 transcriptional regulatory protein leads to reactivation of the dormant p53 and p105 (Rb) tumor suppressor pathways, telomerase inhibition, and deep growth arrest [152]. Irradiation of immortalized cells with ^60^Co gamma radiation increased the mutated *p53* gene and decreased the *sdiI/p21* and *MDM2* expressions [153]. The *sdiI/p21* gene significantly reduces the DNA synthesis capacity, indicating that the *p53* cascade may play an important role in human cell immortalization [153]. The promoter methylation of p21 (Waf1/Cip1) is implicated in the aging process by directly repressing its expression in middle-aged fibroblasts and blocking the radiation-induced DNA damage signaling pathway by p53 [154]. Ku80 loss results in increased susceptibility to ionizing radiation, and replicative senescence in ku80(-/-) cells is caused by a p53-dependent cell cycle response to the damaged DNA [155]. In normal fibroblasts, ATM-dependent signaling pathways in response to ionizing radiation-induced DNA damage are not required for the activation of p38 MAPK or stress-induced premature senescence [156]. Mouse embryo fibroblasts deficient of the c-Jun proto-oncogene (c-Jun-/-MEF) undergo p53-dependent premature senescence in conventional culture [157]. Although CK2α and CK2α’ are involved in senescence, the former was responsible for the progression of radiation-induced stem cell senescence [158]. SOCS1 sufficiently induced p53-dependent senescence in fibroblasts after irradiation [159]. Nerve-injury-induced protein 1 (Ninjurin1, Ninj1) is a target of p53 and causes *p53* mRNA translation and p53-dependent premature aging in vitro and in vivo [160]. Insulin-like growth factor I (IGF-1) is a key regulator of IR-induced accelerated aging, requiring a pathway upstream of both p53 and p21/waf1 that requires intact mammalian target of rapamycin (mTOR) activity [161].

## 9. p53 and Senescence Induced by Radiation in Cancer Cells

Radiation-induced senescence is observed in cancer cells via the *p53* gene; however, p53-null lung cancer cell line H1299 has radiation-induced senescence due to p21waf overexpression [162,163]. Furthermore, the overexpression of this cell surface proteoglycan syndecan 1 (SDC1) in senescent cells after irradiation of human breast cancer is the result of autocrine action in TGF-β via the Smad pathway and the transcription factor Sp1 and does not involve classical aging pathways, such as p53 or p38 MAPK-NF-kB [164]. Overexpression of thioredoxin-1 (TRX), which is observed in many primary human cancers, in HT-1080 fibrosarcoma cells, increased the cyclin D1, p53, and p21 expression, with cyclin D1 overexpression being particularly and directly responsible for increased cell senescence and radiosensitivity [165]. In response to fractionated radiotherapy, glioblastoma multiforme cell lines showed senescence in wild-type p53 cells but not in mutant p53 cells [166]. In breast cancer cells treated by 17-beta-estradiol after radiation injury, it does not affect p53 activation but decreases cyclin E binding to p21(waf1/cip1), and downstream Rb hyperphosphorylation persists due to functional p21(waf1/cip1) inactivation, resulting in continued senescence [167]. In MCF-7 mammospheric cells, higher expression of the post-irradiation DNA single-strand break repair (SSBR) protein human AP endonuclease 1 (Ape1) results in a more active SSBR pathway, higher telomerase activity, lower p21 protein expression, and a significantly reduced senescence tendency [168]. In human breast cancer MCF7 cells, low doses of radiation inhibit doxorubicin-induced replicative senescence by suppressing p38-dependent p53 phosphorylation and activating ERK1/2, without genomic damage [169]. In prostate cancer cells, Nutlin-3 activates p53, making it an effective radiosensitizer—an effect completely attributable to the increased induction of p53-dependent cellular senescence [170]. Laryngeal squamous cell carcinoma cells with wild-type p53 showed significantly increased radiosensitivity, cell cycle arrest, and senescence by Nutlin-3 [171]. In human non-small cell lung cancer cells [172] and esophageal squamous cancer [173], Nutlin-3 activates p53 in senescence. Breast cancer cell irradiation induced 53BP1 and poly (ADP-ribose) polymerase inhibitor-accelerated senescence [174]. Specific matrix metalloproteases (MMPs) inhibitors markedly inhibit the growth of malignant lung epithelial cells in the presence of aging fibroblasts [175]. DNAPK inhibition sufficiently causes accelerated aging in irradiated human cancer cells [176]. Regulated in development and DNA damage responses (REDD1) is dramatically suppressed when the NF-κB or *p53* genes are deleted, making it a protective factor in radiation-induced osteoblast premature senescence [177]. The C-X-C motif chemokine receptor (CXCR) 2 is transactivated by p53 and is a major mediator of oncogene-induced senescence [178]. Allosteric inhibitors of the CXCR4 receptor block metastasis-associated cell activity stimulated by secretions from irradiated lung epithelial cells and may be effective as cancer therapy when combined with radiation therapy [179]. Treatment for cancer cells with imidazoacridinone C-1311 (SymadexTM) prior to irradiation causes senescence in the presence of p53 and apoptosis without p53 [180]. A combination of radiotherapy and drug therapy is also important. Irradiation of p53 wild-type non-small cell lung cancer cells in vitro may induce an senescence-associated secretory phenotype (SASP), which expressed CD63 and generates extracellular vesicles with DNA:RNA hybrids and LINE-1 retrotransposons, causing an abscopal phenomenon [181].

## 10. p53 in the Radioadaptive Response (RAR)

The radioadaptive response (RAR) is a phenomenon in which low-dose radiation is administered in advance to induce resistance to subsequent high-dose radiation [182]. In 1984, Olivieri first reported the phenomenon of the radiation adaptive response in vitro [182]. The *p53* gene plays an important role in RAR [183,184,185,186]; however, it has reported that in some cases RAR occur without dependence on p53. [187]. One mechanism of RAR is associated with survivin [188]. In mammalian cells, priming doses of less than 100 mGy have been used [189,190]; however, in vivo studies have reported higher priming doses of 150–600 mGy [191], 300 mGy [192], and 1 Gy [193]. RAR is observed at a priming dose of 20 mGy in the presence of p53 [184]. We have not yet published data in the presence of p53, a priming dose of 20 mGy followed by a 3 Gy extended lifespan more than 3 Gy alone.

RAR has also been observed in reduced fetal malformations [192,194,195,196,197,198,199,200], in which p53 plays an important role [194,196,200]. RARs are found in normal cells but not in tumor cells [201]. In other words, RAR does not occur when the *p53* gene is abnormal [201]. Interestingly, in the RAR in mouse thymocytes, apoptosis-related expression was lower in males than in females [202]. The reason for this finding is that, in males, phosphoserine-389-TRP53 mediates caspase-3, whereas, in females, there is activation of phosphoserine-18-TRP53 and caspase-3 as well as overexpression of Dlc1 and Fis1 [202]. In cells expressing p53, low-dose irradiation with high-LET radiation has been shown to cause RAR, which is characterized by a low mutation frequency at the hypoxanthine-guanine phosphoribosyl transferase locus and a fast DNA repair rate [203]. In individuals living in areas with high natural radiation levels (>5.0 mGy/year), the p53 pathway (CDKN1A (p21), MDM2, TNF receptor associated factor 4 (TRAF4), TNF receptor superfamily member 10b (TNFRSF10B), apoptotic peptidase activating factor 1 (APAF1), phorbol-12-myristate-13-acetate-induced protein (PMAIP1), GADD45B, etc.) is overexpressed in the blood, which explains the phenomenon of RAR. Immune responses, cell–cell communication, and gap junctions also play important roles in RAR [204]. Ligase1 and p53 levels in the blood of operating room personnel are higher than those in the control group due to the anesthetic gas, and, when blood samples from this group were irradiated, the response differed from that of the control group. These results are also not limited to radiation, as exposure to the low concentrations of chemicals can cause RAR, which may be one of the biological defenses [205]. Guéguen Y. et al. provided a detailed review of the adaptive response [206].

## 11. p53 as a Biomarker after Irradiation

Measurement of p53 protein expression or p53 antibodies in the serum was potentially useful for the early detection of lung cancer [207,208,209,210]. The mRNA regulation and metabolism by p53 in response to radiation exposure has identified important mRNAs and metabolites as potential p53 targets [211]. After irradiation with 4 Gy, 326 mRNAs were significantly altered, i.e., 269 and 57 were increased and decreased, respectively [211].

Although p53 expression increased with age [52], it is not a suitable parameter for the early detection of lung cancer in former uranium miners [212]. When human peripheral blood was irradiated with X-rays, genes associated with chemokines (*platelet factor 4 [PF4]*, *G protein subunit gamma 11 [GNG11]* and *C-C motif chemokine receptor 4 [CCR4]*) were independently expressed from the *p53* gene at 0.05 Gy, whereas genes involved in *p53* gene signaling (*damage-specific DNA binding protein 2 [DDB2]*, *apoptosis enhancing nuclease [AEN]* and *CDKN1A [p21]*) were expressed at 1 Gy [213].

Furthermore, 147 metabolites were significantly expressed: 45 were increased and 102 were decreased. Particularly, gamma-glutamyl transferase 1 [GGT1], Phospholipase A2G [PLA2], post-transcriptional gene silencing [PTGS], glutathione peroxidase 6 [GPX6], aldolase, fructose-bisphosphate A [ALDOA], acyl-CoA synthetase short chain family member 2 [ACSS2], aldehyde dehydrogenase 3 family member A1 [ALDH3A1], and gamma-glutamyl transferase 6 [GGT6], involved in nitrogen metabolism, glutathione metabolism, glycolysis, or glycogenesis, and arachidonic acid metabolism, were suggested to be regulated by p53 [211]. DDB2, AEN, TP53 regulated inhibitor of apoptosis (TRIAP1), and TRAF4, which are all downstream of p53, are potential biomarkers within the first 24 h after radiation exposure [214].

## 12. Conclusions

The roles of p53 isoforms are still unknown, although they have already been identified. Specifically, only a few reports demonstrated the association between radiation and isoforms.

This review described the association between p53 and radiation. MDM2 has already been known to regulate p53. Regarding the NF-κB, several reports revealed that it may be p53-dependent. Although miRs are expected as biomarkers, various types and roles of miRs are still unclear. Although miR34 and let-7 have been widely associated with p53 and miRs after irradiation, their association with other miRs types is also diverse.

Inflammatory findings persist after radiation exposure in atomic bomb survivors [215,216]. Moreover, inflammation is a cause of carcinogenesis [105]. Senescence after radiation exposure is observed in normal and cancer cells and occurs when the cell cycle is stopped. Therefore, the role of p53 in these phenomena is significant.

This review described the role of p53 in radiation. The number of publications on p53, including those on DNA damage due to various mutagenic agents, is too numerous to mention even for related genes. p53 has many roles and continues to play various roles in the future. Hopefully, this review will contribute to further research.

## Figures and Tables

**Figure 1 life-12-01099-f001:**
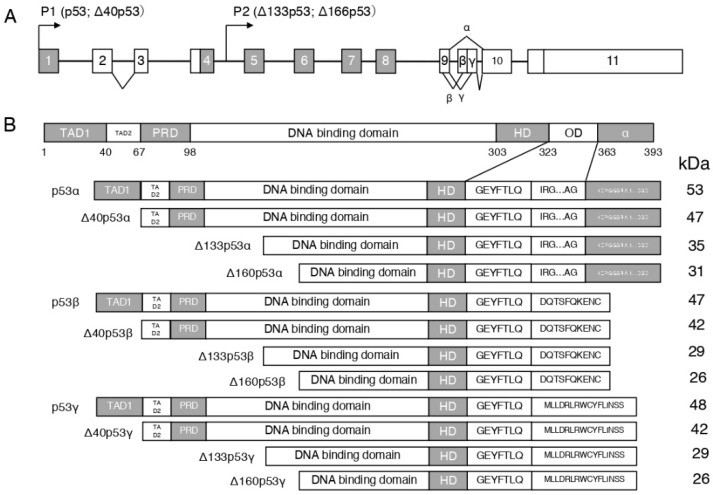
Schema of the human *p53* gene structure: (**A**): alternative splicing sites (α, β, and γ) and promoters (P1 and P2) are marked. (**B**): Schema of the human p53 protein isoforms can be expressed by the human *p53* gene. TAD, transcription activation domain; PRD, proline-rich domain; HD, hinge domain; and OD, oligomerization domain.

**Figure 2 life-12-01099-f002:**
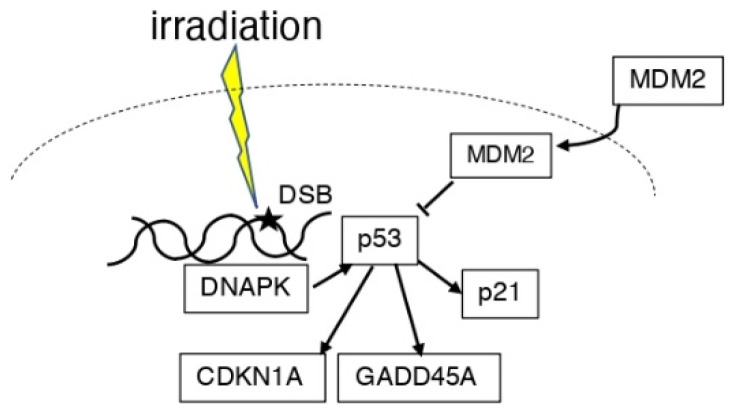
Scheme of the association between p53 and MDM2 in irradiation. DNA damage induced by ionizing radiation induces p53. DNAPK can also induce p53 expression. MDM2 is present in the cytoplasm and nucleus and suppresses p53 function. P53 expresses p21, GADD45A, and CDKN1A.

**Figure 3 life-12-01099-f003:**
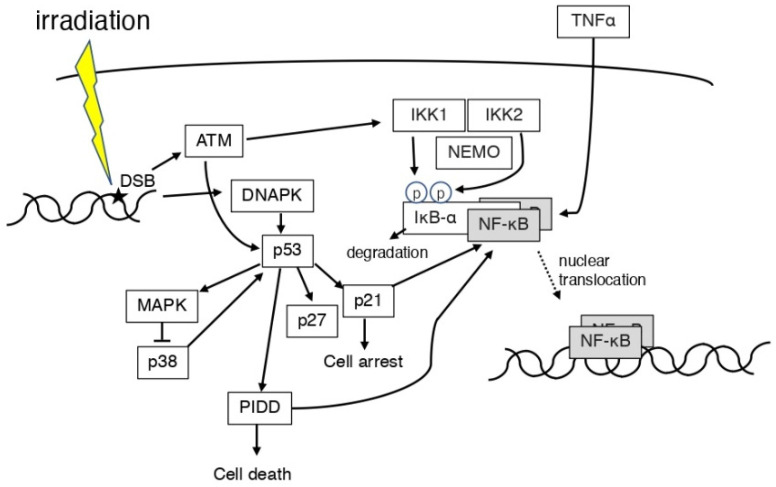
Scheme of the association between p53 and NF-κB in irradiation. ATM and DNAPK are expressed by ionizing radiation causing DNA damage. PIDD is involved in the NF-κB expression. ROS and ATM induced IKK1 and IKK2. IκB-α degrades and activates NF-κB. TNFα may also activate NF-κB. p53 and NF-κB may be dependent or independent.

**Figure 4 life-12-01099-f004:**
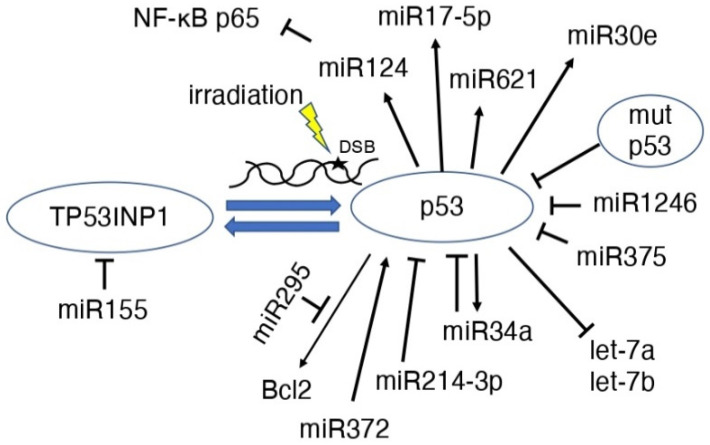
Scheme of the association between p53 and miRNAs in irradiation MiRNAs induce p53 or suppress p53 function.

## Data Availability

Not applicable.

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
