# Peer review of "Role of p53 in Regulating Radiation Responses"

_life, 2022, doi:10.3390/life12071099_

Round 1

Reviewer 1 Report

In this review the author describes the role of p53 in the response to gamma irradiation through its interaction with MDM2, NF-κB and miRNA, as well as in the processes of inflammation, senescence and carcinogenesis in general.

The review is not well organized and there are many misleading information.

I have a feeling that the author is not an expert in the field.

The abstract is not properly written, it lacks the information on the manuscript. Needs to be rewritten.

p53 isoforms and their meaning are not mentioned.

Table 1 is not informative at all.

Many misleading facts, e.g. lanes 81-82: “s, low-dose fractionated irradiation (LDFRT, <1 Gy) reduced NF-κB activity and increased BAX protein expressing, corresponding to an antiapoptotic effect” is false!

More figures would help to understand.

I cannot agree that “MAPK pathway molecules, p38, p53, p21, and p27”, lane 91 although it is published.

Author Response

In this review the author describes the role of p53 in the response to gamma irradiation through its interaction with MDM2, NF-κB and miRNA, as well as in the processes of inflammation, senescence and carcinogenesis in general.The review is not well organized and there are many misleading information.I have a feeling that the author is not an expert in the field.

I appreciate your constructive comments. With that, I have corrected the areas pointed out. I hope the response to your comments is satisfactory.

The abstract is not properly written, it lacks the information on the manuscript. Needs to be rewritten.

I agreed with you. I rewrote new abstract in new version as follows (line 14-23);

p53 is known as the guardian of the genome and has many roles in DNA damage and cancer suppression. The TP53 gene was found to express multiple p53 splice variants (isoforms) in a physiological, tissue-dependent manner. Various genes upstream and downstream of p53 are involved in cell viability, senescence, inflammation, and carcinogenesis. And also, the p53 gene plays an important role in radioadaptive response. p53 is important in radiation research, and many research papers have been published. In this review, the role of p53 in the response to gamma irradiation through its interaction with MDM2, NF-κB and miRNA, as well as in the processes of inflammation, senescence, carcinogenesis and radiation adaptive responses in general will be presented. Finally, the potential of p53 as a biomarker was discussed.

p53 isoforms and their meaning are not mentioned.

In the revised paper, I presented a description of p53 isoform and several papers on its radiation effects as follows (line 51-70);

p53 isoforms and its response to radiation (Figure 1).

The TP53 gene is composed of 11 exons [25,26]. Human TP53 has been reported to express at least nine mRNAs in normal tissues in a tissue-dependent manner. These are the result of alternative usage of promoters (P1 and P2) and alternative splicing of intron-2 and intron-9. There are 12 isoforms of p53 protein: p53α, p53β, p53γ, Δ40p53α, Δ40p53β, Δ40p53γ, Δ133p53α, Δ133p53β, Δ133p53γ, Δ160p53α, Δ160p53 β, and Δ160p53γ have been reported. Depending on the cell type, translation of p53 mRNA starts at different codons. mRNA transcribed from the proximal promoter (P1) initiates translation at codon 1 and/or 40, while mRNA transcribed from the internal promoter (P2) initiates translation at codon 133 and/or 160.

By irradiation, the p53 isoform Δ113p53 is a p53 target gene that antagonizes the apoptotic activity of p53 via activation of BclxL in zebrafish [27]. Restoration of ∆133p53 expression protects astrocytes from radiation-induced senescence, promotes DNA repair, and inhibits astrocyte-mediated neuroinflammation [28]. Different radiation doses in germ cells during Drosophila oogenesis play different roles depending on p53 isoforms (p53A and p53B) [29]. In the absence of stress, p53A is expressed to support normal oogenesis, and in the early stages of oogenesis, p53A and p53B are involved in radiation-induced apoptosis, and both isoforms were involved in restoring the number of germline stem cells after high dose of irradiation [29]. p53A, but not p53B, was involved in the maintenance of germline stem cells function after high doses of irradiation [29]

Table 1 is not informative at all.

I agreed with you. Table 1 remove in new version, and added new figure.

Many misleading facts, e.g. lanes 81-82: “s, low-dose fractionated irradiation (LDFRT, <1 Gy) reduced NF-κB activity and increased BAX protein expressing, corresponding to an antiapoptotic effect” is false!

I deleted this sentence.

More figures would help to understand.

I made three more figures in new version.

I cannot agree that “MAPK pathway molecules, p38, p53, p21, and p27”, lane 91 although it is published.

I deleted this sentence. 

Reviewer 2 Report

After reading this review, my personal opinion is that the author is a radiation field research but not so familiar with p53 field. Author should cooperate with one p53 field expert and rewrite this review to reconsider to publish this review paper.

Other comments:

In line 28:

cell cycle arrest (p21, 14-3-3d, Riprimo, GADD45, mir34, CDK2,CCNE1, etc.)

14-3-3d should be 14-3-3 σ, and mir34 should be miR-34.

In line 57:

Radiation-induced DNA damage activates p53, and MDM2 can stabilize or activate p53 [35,36]. => this sentence is totally wrong, MDM2 is never to activate p53.

In addition, Reference 35 and 36 is not related to Radiation, and two references just told that MDM2 how to inhibit p53 function.

Line 72-105 were mentied that the crosslink of NfKB and p53 which was so confused and hard to understand, and table 1 is too far away form main article and readers cannot get useful information about  table 1.

Figure 1. has no figure legend.

Line 123-126:

The expression of let-7a and let-7b, which are downstream of p53, was downregulated by radiation and corelated with changes in the expression of proteins in the p53-regulated apoptosis promoting signaling pathway, which are important for radiation-induced cytotoxicity.

"downstream of p53 " is general mentioned as the p53 direct binds and activates genes, but let-7a and let-7b were repressed by p53.

Line 140-141

Increased miR-372 activated p53 via suppression of the PDZ-binding kinase and increased radiosensitivity of tumor cells [76].

Reference 76 is mentioned miR-372 activate p53, but Fig1 draw p53 activate miR-372 which totally conflict. 

Author Response

After reading this review, my personal opinion is that the author is a radiation field research but not so familiar with p53 field. Author should cooperate with one p53 field expert and rewrite this review to reconsider to publish this review paper.

I appreciate your constructive comments. With that, I have corrected the areas pointed out.

In writing this review, I was asked to write about the role of p53 in radiation. Therefore, it is not specific to p53 in general. The title is also “Role of p53 in Regulating Radiation Responses”. I hope the response to your comments is satisfactory.

Other comments:

In line 28:

cell cycle arrest (p21, 14-3-3d, Riprimo, GADD45, mir34, CDK2,CCNE1, etc.)

14-3-3d should be 14-3-3 σ, and mir34 should be miR-34.

I corrected. (line 36-37)

In line 57:

Radiation-induced DNA damage activates p53, and MDM2 can stabilize or activate p53 [35,36]. => this sentence is totally wrong, MDM2 is never to activate p53.

In addition, Reference 35 and 36 is not related to radiation, and two references just told that MDM2 how to inhibit p53 function.

I deleted this sentence.

Line 72-105 were mentied that the crosslink of NFkB and p53 which was so confused and hard to understand, and table 1 is too far away form main article and readers cannot get useful information about table 1.

I agreed with you. Table 1 remove in new version.

Figure 1. has no figure legend.

I added figure legend.

Line 123-126:

The expression of let-7a and let-7b, which are downstream of p53, was downregulated by radiation and corelated with changes in the expression of proteins in the p53-regulated apoptosis promoting signaling pathway, which are important for radiation-induced cytotoxicity.

"downstream of p53 " is general mentioned as the p53 direct binds and activates genes, but let-7a and let-7b were repressed by p53.

I added the sentence as follows (line 140-141);

In other words, the p53 direct binds and activates genes, but let-7a and let-7b were repressed by p53. 

Line 140-141

Increased miR-372 activated p53 via suppression of the PDZ-binding kinase and increased radiosensitivity of tumor cells [76].

Reference 76 is mentioned miR-372 activate p53, but Fig1 draw p53 activate miR-372 which totally conflict. 

I agreed with you. I changed in new Figure 4

Reviewer 3 Report

The article titled “Role of p53 in regulating radiation responses” by Okazaki R offers an organized resource for a protein that has been the subject of heavy research for several decades. The scope of the manuscript has been appropriately limited to the topic of radiation biology rather than try to encompass multiple biological responses for which p53 serves as master regulator. The following comments may enhance the readability of the manuscript and increase likelihood of citations:

1)      The association between p53 and different microRNAs described on page 3 is hard to follow. This information may be best represented in a tabular format. The portion on mir-34a reads better than the rest of the text

2)      Lines 39-52: The paragraph needs to have a core message. The title of the paragraph is also misleading. The text in this paragraph is not describing how radiation causes DNA damage. IT is explaining loss of heterozygosity. Also it is not clear what role of p53 the author intends to highlight here.

3)      Line 67- “we found” should be changed to “it was found”

4)      Line 315, I assume author meant to say: In breast cancer cells “treated” by?

5)      Line 358-359 is the author describing own work here?

6)      The section on p53 as a biomarker after irradiation needs to be restructured for clarity. It can be divided into 2 paragraphs. One describing contexts where it could be a biomarker and another for examples where it does not.

Author Response

The article titled “Role of p53 in regulating radiation responses” by Okazaki R offers an organized resource for a protein that has been the subject of heavy research for several decades. The scope of the manuscript has been appropriately limited to the topic of radiation biology rather than try to encompass multiple biological responses for which p53 serves as master regulator. The following comments may enhance the readability of the manuscript and increase likelihood of citations:

I appreciate your constructive comments. With that, I have corrected the areas pointed out. I hope the response to your comments is satisfactory.

1) The association between p53 and different microRNAs described on page 3 is hard to follow. This information may be best represented in a tabular format. The portion on mir-34a reads better than the rest of the text.

I have divided the description of microRNAs and p53 into three paragraphs: miR34 and let and other mRNAs.

There was a mistake in the diagram (arrow direction for miR-372), which has been corrected.

2) Lines 39-52: The paragraph needs to have a core message. The title of the paragraph is also misleading. The text in this paragraph is not describing how radiation causes DNA damage. IT is explaining loss of heterozygosity. Also, it is not clear what role of p53 the author intends to highlight here.

I agreed with you. I deleted this sentence because it does not make good sense.

3)      Line 67- “we found” should be changed to “it was found”

I agreed with you. I corrected. (line 84)

4)      Line 315, I assume author meant to say: In breast cancer cells “treated” by?

I agreed with you. I corrected. (line 338)

5)      Line 358-359 is the author describing own work here?

Yes, it is. I changed the sentence as follows;

We have not yet published data, in the presence of p53, a priming dose of 20 mGy followed by 3 Gy extended lifespan more than 3 Gy alone. (line 384-385)

6)      The section on p53 as a biomarker after irradiation needs to be restructured for clarity. It can be divided into 2 paragraphs. One describing contexts where it could be a biomarker and another for examples where it does not.

I agreed with you. I revised as follows (408-425);

Measurement of p53 protein expression or p53 antibodies in serum was suggested to be potentially useful for early detection of lung cancer [203,204,205,206]. The regulation of mRNA and metabolism by p53 in response to radiation exposure has identified important mRNAs and metabolites as potential targets of p53 [207]. After irradiation with 4 Gy, 326 mRNAs were significantly altered: 269 mRNAs were increased and 57 were decreased [207].

Although p53 expression increased with age [47], it is not a suitable parameter for early detection of lung cancer in former uranium miners [208]. When human peripheral blood was irradiated with X-rays, genes related to chemokines (PF4, GNG11 and CCR4) were expressed independently of the p53 gene at 0.05 Gy, while genes involved in p53 gene signaling (DDB2, AEN and CDKN1A) were expressed at 1 Gy [209].

In addition, 147 metabolites were significantly: 45 were increased and 102 were decreased. In particular, GGT1, PLA2G, PTGS, GPX6, ALDOA, ACSS2, ALDH3A1, and GGT6, which are involved in nitrogen metabolism, glutathione metabolism, glycolysis or glycogenesis, and arachidonic acid metabolism, were suggested to be regulated by p53 [207]. DDB2, AEN, TRIAP1, and TRAF4, which are downstream of p53, are potential biomarkers within the first 24 hours after radiation exposure [210].

Reviewer 4 Report

In this review, author demonstrated role of p53 in regulating radiation responses. Many previous papers have been introduced and it is a useful review for readers. However, I request the author to add a visual figure for each section to help the reader's understanding. In addition, I think it is appropriate to delete Table 1 because the explanation in the text is sufficient. 

Author Response

I appreciate your constructive comments. With that, I have corrected the areas pointed out. I agreed with you. I deleted table 1. Instead, I added figures. I hope the response to your comments is satisfactory.

Round 2

Reviewer 1 Report

Abstract: “And also, the p53 gene plays 11 an important role in radioadaptive response. p53 is important in radiation research, and many re-12 search papers have been published.“ - These two sentences are very similar! Merge them into one, please.

The sentence stating “MAPK pathway molecules, p38, p53, p21, and p27” is still there, twice, lines 100 and 192.

The Conclusion has to be rewritten.

There is inconsistency in writing TP53 gene. p53 is usually used for protein.

English has to be improved!

Author Response

Abstract: “And also, the p53 gene plays 11 an important role in radioadaptive response. p53 is important in radiation research, and many re-12 search papers have been published.“ - These two sentences are very similar! Merge them into one, please.

I changed the sentence as follows (line 17)

Numerous research papers have been published on p53.

The sentence stating “MAPK pathway molecules, p38, p53, p21, and p27” is still there, twice, lines 100 and 192.

Thank you for pointing out the double sentence. The latter sentence has been deleted. The Conclusion has to be rewritten.

There is inconsistency in writing TP53 gene. p53 is usually used for protein.

Thanks for pointing this out. The p53 indicating the gene has been changed to italics.

English has to be improved!

Corrections have been made to the English language as noted.

Reviewer 2 Report

In this revised version, I still think the author is not preparing so well.

1.      All figure legends were absent.

2.      In line 46: Human TP53 has been reported to express at least nine mRNAs in normal tissues in a tissue-dependent manner. (Ref??)

3.      In line 55-56: By irradiation, the p53 isoform Δ113p53 is a p53 target gene that antagonizes the apoptotic activity of p53 via activation of BclxL in zebrafish [27].

One p53 isoform is human Δ133p53 and its zebrafish counterpart Δ113p53.  bcl2L (closest to human Bcl-xL)

So it is needed to correct as following:

By irradiation, the p53 isoform Δ113p53 (as Δ133p53 in human) is a p53 target gene that antagonizes the apoptotic activity of p53 via activation of bcl2L (as Bcl-xL in human ) in zebrafish [27].

4.      For line55-64, author mentioned all the irradiation response in non-human model (in zebrafish and Drosophila). Drosophila p53 isoforms is big difference with human. So the information about p53 isoforms in Drosophila may not so helpful in here. In addition, it had been known that p63, but not p53 involves in irradiation mediated response in human oocyte (eLife. 2016; 5: e13909).  

In line 77-78 and line 81 mdm2 should be MDM2 as previous mentioned.

In line 67-83, many molecules were not mentioned but appeared in Fig2, such as GADD45, CDK1 and cyclin1.

In Fig2, why need to draw the both CDK1/cyclin1 and CDK/cyclin. It’s look like redundant. Lightning icon in Fig2 is what kind of meaning? X-ray? Ultraviolet? DNA damage? The star icon in Fig2 is what kind of meaning? The same problems also appeared in Fig3.

In line 85-114, many molecules were not mentioned but appeared in Fig3, such as PIDD, ROS, IKK1, IKK2, NEMO and cyclin1.

In Fig4, it had no information about irradiation. The reader just can get the relationship about microRNA and p53.

In line 117-159, many molecules were not mentioned but appeared in Fig4, such as SIRT1, HDAC1, COP1, and MDM4.

Author Response

In this revised version, I still think the author is not preparing so well. 

  1. All figure legends were absent. 

I added legends in all figures.

Figure 1. Schema of the human p53 gene structure.

A: alternative splicing sites (α, β, γ) and promoters (P1, P2) are marked. B: Schema of the human p53 protein isoforms can be expressed by the human p53 gene. TAD = transcription activation domain, PRD = proline-rich domain, HD = Hinge domain, OD = oligomerization domain.

Figure 2. Scheme of association between p53 and MDM2 in irradiation

DNA damage induced by ionizing radiation induces p53. DNAPK can also induce p53 expression. MDM2 is present in the cytoplasm and nucleus and suppresses p53 function. p53 expresses p21, GADD45A and CDKN1A

Figure 3. Scheme of association between p53 and NF-κB in irradiation

ATM, and DNAPK are expressed by ionizing radiation causing DNA damage. p53 is induced by ATM and DNAPK. p53 induces MAPK, p21, and PIDD. PIDD is involved in the expression of NF-κB. ROS and ATM induce IKK1 and IKK2. IκB-α degrades and activates NF-κB. TNFα may also activate NF-κB. p53 and NF-κB may be dependent or independent.

Figure 4. Scheme of association between p53 and miRNAs in irradiation
MiRNAs induce p53 or suppress p53 function.

  1. In line 46: Human TP53 has been reported to express at least nine mRNAs in normal tissues in a tissue-dependent manner. (Ref??)

This sentence has been deleted because the source is no longer known.

More information on the p53 protein has been added (line 55-62).

The TP53 gene is composed of 11 exons [25,26]. These are the result of alternative usage of promoters (P1 and P2) and alternative splicing of intron-2 and intron-9 (Figure 1A). The p53 protein is composed of 393 amino acids and is classified into six domains (Figure 1B) [25,27]. The p53 protein consists of 393 amino acids and is divided into six domains: Transcriptional Activation Domain (TAD) (residues 1-67), TAD I (residues 1-40) and TAD II (residues 41-67): interact with a variety of proteins; the Proline-rich region (residues 67-98): conserved in most p53; central core domain (residues 98-303): DNA-binding domain where more than 90% of p53 mutations causing cancer in humans are found; nuclear localization signal located at residues 303-323; tetramerization domain at residues 323-363 domain, C-terminal basic domain (residues 363-393): nonspecific DNA binding domain that recognizes and binds damaged DNA [25,27].

  1. In line 55-56: By irradiation, the p53 isoform Δ113p53 is a p53 target gene that antagonizes the apoptotic activity of p53 via activation of BclxL in zebrafish [27].

One p53 isoform is human Δ133p53 and its zebrafish counterpart Δ113p53.  bcl2L (closest to human Bcl-xL)

So it is needed to correct as following:

By irradiation, the p53 isoform Δ113p53 (as Δ133p53 in human) is a p53 target gene that antagonizes the apoptotic activity of p53 via activation of bcl2L (as Bcl-xL in human ) in zebrafish [27].

I agreed with you. Thank you for your correcting it. (line 69-71)

  1. For line55-64, author mentioned all the irradiation response in non-human model (in zebrafish and Drosophila). Drosophila p53 isoforms is big difference with human. So the information about p53 isoforms in Drosophila may not so helpful in here. In addition, it had been known that p63, but not p53 involves in irradiation mediated response in human oocyte (eLife. 2016; 5: e13909).  

In line 77-78 and line 81 mdm2 should be MDM2 as previous mentioned.

I agree with you. I changed mdm2 toMDM2.

  • In line 67-83, many molecules were not mentioned but appeared in Fig2, such as GADD45, CDK1 and cyclin1. 

I agree with you. Anything not in the text has been removed in the figure.

  • In Fig2, why need to draw the both CDK1/cyclin1 and CDK/cyclin. It’s look like redundant. Lightning icon in Fig2 is what kind of meaning? X-ray? Ultraviolet? DNA damage? The star icon in Fig2 is what kind of meaning? The same problems also appeared in Fig3. 

Lightning icon shows radiation. The star icon shows DSB. In all figures, the words are added.

  • In line 85-114, many molecules were not mentioned but appeared in Fig3, such as PIDD, ROS, IKK1, IKK2, NEMO and cyclin1. 

I agree with you. Anything not in the text has been removed in the figure. Sentences have been added to the text regarding PIDD, IKK1, IKK2 and NEMO. (line109, 121-123)

  • In Fig4, it had no information about irradiation. The reader just can get the relationship about microRNA and p53. 

I agree with you. I added information about irradiation in Fug. 4.

  • In line 117-159, many molecules were not mentioned but appeared in Fig4, such as SIRT1, HDAC1, COP1, and MDM4. 

I agree with you. Anything not in the text has been removed in the figure.

Reviewer 3 Report

Minor langue corrections and rephrasing needed at various points throughout the text 

Author Response

Minor langue corrections and rephrasing needed at various points throughout the text 

Corrections have been made to the English language as noted.